# Don't Stop Me Now:
# Embedding Based Scheduling for LLMs

**Rana Shahout**
Harvard University

**Eran Malach**
Harvard University

**Chunwei Liu**
MIT

**Weifan Jiang**
Harvard University

**Minlan Yu**
Harvard University

**Michael Mitzenmacher**
Harvard University

## Abstract

Efficient scheduling is crucial for interactive Large Language Model (LLM) applications, where low request completion time directly impacts user engagement. Size-based scheduling algorithms like Shortest Remaining Process Time (SRPT) aim to reduce average request completion time by leveraging known or estimated request sizes and allowing preemption by incoming requests with shorter service times. However, two main challenges arise when applying size-based scheduling to LLM systems. First, accurately predicting output lengths from prompts is challenging and often resource-intensive, making it impractical for many systems. As a result, the state-of-the-art LLM systems default to first-come, first-served scheduling, which can lead to head-of-line blocking and reduced system efficiency. Second, preemption introduces extra memory overhead to LLM systems as they must maintain intermediate states for unfinished (preempted) requests.

In this paper, we propose TRAIL, a novel approach that improves the response time in LLM inference through two key contributions: (1) obtain output predictions from the target LLM itself. After generating each output token, we recycle the embedding of its internal structure as input for a lightweight classifier that predicts the remaining length for each running request. (2) prediction-based SRPT variant with limited preemption designed to account for memory overhead in LLM systems. This variant allows preemption early in request execution when memory consumption is low but restricts preemption as requests approach completion to optimize resource utilization. On the theoretical side, we derive a closed-form formula for this SRPT variant in an M/G/1 queue model, which demonstrates its potential value. In our system, we implement this preemption policy alongside our embedding-based prediction method. Our refined predictions from layer embeddings achieve 2.66x lower mean absolute error compared to BERT predictions from sequence prompts. TRAIL achieves 1.66x to 2.01x lower mean latency on the Alpaca dataset and 1.76x to 24.07x lower mean time to the first token compared to the state-of-the-art serving system.

## 1 Introduction

Recent advances in large language models (LLMs) have sparked a new generation of interactive AI applications. OpenAI's ChatGPT (OpenAI, 2022) exemplifies this trend, facilitating conversational interaction across diverse tasks. The interactive nature of these applications requires low request completion time to ensure user engagement. Users expect near-instant responses, making efficient inference serving critical for LLM-based interactive AI applications.

LLM inference requests exhibit a distinct autoregressive pattern comprising multiple iterations in which each output token is appended to the input to generate the next token. Orca (Yu et al., 2022) and vLLM (Kwon et al., 2023) are state-of-the-art solutions for LLM inference systems. These systems use iteration-level scheduling, which allows new requests to be added or completed requests

to be removed at the end of each iteration, providing greater flexibility in managing the processing batch than request-level scheduling. However, these works still process requests using a first-come-first-served (FCFS) policy, leading to head-of-line blocking (Kaffes et al., 2019) under high load. This problem is particularly severe for LLM inference, where many short requests may wait for other long requests to finish.

Size-based scheduling aims to reduce request completion time by leveraging known or estimated request sizes. For LLM requests, the execution time depends on both input and output lengths, with the latter being unknown and potentially variable. Preemption allows for dynamic scheduling: when one scheduled request finishes generating an output token, we can decide whether to continue this request or preempt it with another (newly arrived) request. The Shortest Remaining Process Time (SRPT) policy is an example of preemption scheduling. However, preemption here introduces extra memory overhead to maintain an intermediate state for started but unfinished requests. LLMs maintain a key-value (KV) cache for each Transformer layer to store the intermediate state. In preemption scheduling policies, the cache must keep the intermediate state for all started but unfinished requests in the waiting queue, which may cause memory overflow, given the large size of LLMs and the limited memory capacity of GPUs. (Non-preemptive policies, e.g. FCFS, do not face this issue, as there are no unfinished requests in the waiting queue.)

Recent works propose methods to predict request sizes. $S^3$ (Jin et al., 2023) fine-tunes a BERT model (Sanh, 2019) to predict output sequence lengths from input prompts. While this model is lightweight in resources, its accuracy diminishes for requests with varying execution times. Zheng et al. (Zheng et al., 2024) employ a separate, lighter LLM to predict output lengths before scheduling, achieving higher precision than the BERT model. However, this approach consumes more resources, raising questions about its efficiency and cost-effectiveness in practical applications. Another method proposed by the same work is *Perception in Advance*, where models are asked to predict the length of the responses they are about to generate. However, this approach is limited by potential dependencies between the prediction and the generated output, which may affect the quality of the output.

In this paper, we introduce TRAIL (*Token Response and Embedding Adaptive Inference Layer*), a novel approach that improves the response time in LLM inference through two key contributions: (1) iteration-level predictions for request lengths with low overhead and high accuracy to enable size-based scheduling, and (2) preemption scheduling at token granularity that accounts for memory overhead. For predictions, our approach takes advantage of the autoregressive nature of LLM output generation to predict the output size and facilitate the prediction of request length at the granularity of individual tokens. We begin with an initial prediction based on the prompt and iteratively refine these predictions as the sequence progresses by innovatively using LLM layer embeddings. Specifically, we "recycle" the embedding from intermediate Transformer layers of the LLM, feeding them into a lightweight linear classifier to predict the remaining sequence length. This method, which involves analyzing the outputs of the internal model layer by using them as input for a separate machine learning predictor, is commonly referred to as *probing* (Belinkov, 2022; Hewitt & Liang, 2019; Hewitt & Manning, 2019). This approach combines the advantages of LLM-based prediction with computational efficiency, eliminating the need for an additional LLM dedicated solely to length predictions. To address the memory constraints for preemption scheduling, TRAIL limits the number of times each request can be preempted and introduces SRPT with limited preemptions. Early in the request's execution, preemption is allowed since its KV cache memory usage is small, making preemption less costly in terms of memory. However, as the request progresses and its memory consumption in the KV cache grows, preemption becomes more expensive. To manage this, we turn the preemption off in the later stages of the request to avoid a heavy memory overhead.

Our contributions in this paper are (1) an output length prediction framework using recycled LLM embeddings, with dynamic refinement at each iteration. In our experiments, refined predictions from layer embeddings achieve 2.66x lower mean absolute error compared to BERT predictions from sequence prompts (2) a proposed variant of Shortest Remaining Processing Time using these predictions with limited preemptions (3) a derivation of a closed-form formula for this SRPT variant in an M/G/1 queue model. We also show via experiments that our prediction and scheduling approaches result in mitigating head-of-line blocking. When TRAIL is integrated with a state-of-the-art LLM system, it has 1.66x to 2.01x lower mean latency and 1.76x to 24.07x lower mean time to first token compared to the state-of-the-art serving system tested with the Alpaca dataset. Our

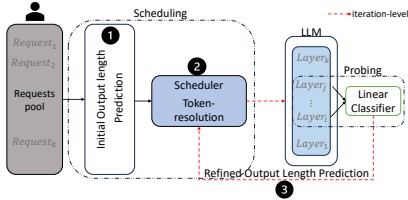

Figure 1: TRAIL architecture. The system (1) initially orders requests using a BERT model, (2) schedules requests using a modified SPRPT with limited preemption, and (3) refines predictions during token generation using embeddings from the LLM's internal layers. At every iteration, steps 2 and 3 are repeated (represented as red dashed lines), which allows preemption at iteration-level granularity and refined predictions. We focus on identifying the LLM layer that best predicts output length rather than using multi-layer embeddings ($i = j = 11$).

TRAIL approach has the potential to significantly enhance LLM inference efficiency and reliability, paving the way for more adaptive LLM serving systems.

## 2 BACKGROUND AND MOTIVATION

**Transformer-Based Generative Models and Key-Value Cache.** At each step, a Transformer model predicts the next most likely token based on the sequence of tokens generated so far. A generative model of length $n$ must perform $n$ iterations, with each token passing through multiple transformer layers composed of self-attention and feed-forward networks.

In a typical iteration at the $i$-th step, the model computes over all previously generated tokens $(t_0, t_1, \ldots, t_{i-1})$ via self-attention. This can be expressed as:

$$h_{\text{out}} = \text{softmax}\left(\frac{q_i \cdot K^{\top}}{\sqrt{d_h}}\right) \cdot V$$

Where $q_i$ is the query vector representing the hidden state of the current token $t_i$, and $K, V \in \mathbb{R}^{i \times d_h}$ are matrices representing the keys and values derived from all previous tokens.

To enhance efficiency, LLMs cache the key and value matrices (KV cache) throughout the sequence generation process, eliminating the need to recompute them at every iteration. This caching mechanism significantly reduces computation time but requires substantial memory proportional to the number of layers and hidden dimensions. As more tokens are generated, the cache stores information from all previously generated tokens, and its memory demand grows linearly with the length of the sequence. This makes the memory consumption of the KV cache considerable, especially for long sequences. For example, in the GPT-3 175B model, a single request with a sequence length of 512 tokens requires at least 2.3 GB of memory to store the key-value pairs. The limited capacity of GPU memory restricts the size of the KV cache and poses a challenge to efficiently implementing preemptive scheduling policies.

**Iteration-Level Scheduling.** Iteration-level scheduling differs from the conventional request-level scheduling approach. In request-level scheduling, the system processes a batch of requests to completion, forcing requests that finish earlier to wait until the entire batch completes, while new requests must remain in a queue until the next batch begins. In contrast, iteration-level scheduling (Yu et al., 2022) offers flexibility by processing only a single iteration (i.e., generating one output token) for each request in the batch. After each iteration, the scheduler is called, and the completed requests can exit the batch while newly arrived requests can be added, allowing dynamic batch adjustments. However, the batch size remains constrained by the GPU memory capacity.

## 3 METHOD

TRAIL leverages the autoregressive nature of LLM inference and iteration-level scheduling, employing two primary strategies to reduce LLM inference response time: iteration-level prediction

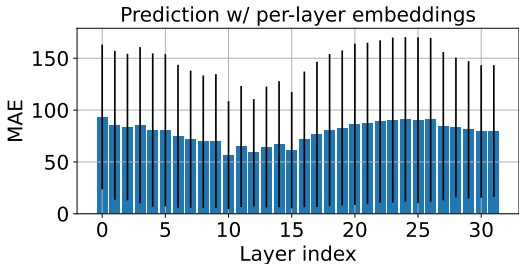

Figure 2: MAE for length prediction using embeddings vs. layer (1,000 prompts).

and preemption that accounts for memory overhead. TRAIL uses low-overhead, high-accuracy predictions by utilizing the intermediate layer embeddings of the LLM itself to predict output length while implementing token-granular, limited preemption scheduling to account for the memory overhead associated with preemption. Figure 1 illustrates the TRAIL architecture. Users submit requests to the request pool. TRAIL initially orders requests using a BERT model based on input prompts (Step 1). This method relies exclusively on the prompt using BERT suggested in (Jin et al., 2023). Given the predictions of the output lengths, the scheduler implements a variant of the Shortest Predicted Remaining Process Time (SPRPT) with limited preemption (Step 2, Section 3.3). Although SPRPT is traditionally designed for single-server, sequential scheduling, we adapt it to handle multiple concurrent requests while still prioritizing those with the shortest predicted remaining time. The number of requests that can be scheduled simultaneously is limited by the available GPU memory. As the LLM generates each output token, we refine predictions using embeddings from the LLM's intermediate layers. A linear classifier processes these embeddings and informs the scheduler, which adjusts based on updated predictions (Section 3.1).

## 3.1 REFINED OUTPUT LENGTH PREDICTION

In this section, we describe our method for estimating the prompt length based on intermediate layer embeddings in the LLM.

***Problem Definition.*** Denote by $\boldsymbol{u}^{(1)}, \ldots, \boldsymbol{u}^{(N)} \in \mathbb{R}^d$ the output of some intermediate layer for the $N$ output tokens generated by the model (where $d$ is the hidden dimension of the Transformer). Additionally, we denote by $\boldsymbol{u}^{(0)}$ the embedding of the input (during the prefilling phase), which is generated by averaging the embeddings of all the input tokens at the given layer. Let $B_1, \ldots, B_k \subset \mathbb{N}$ be a choice of bins for the length prediction, i.e. $B_i = \{b_i, b_i+1, \ldots, b_{i+1}-1\}$ for some choice of bin boundaries $b_1 < b_2 < \cdots < b_{k+1}$. We train a linear classifier that, given some embedding $\boldsymbol{u}^{(t)}$, outputs a vector $p^{(t)} \in [0, 1]^k$ where $p^{(t)}(j)$ indicates the probability that the number of remaining tokens (i.e., $N - t$) is in $B_j$.

Our approach focuses on identifying which LLM layer provides the most accurate predictions for the output length. One potential extension would be to select multiple layers and estimate the prediction using either a weighted average of their outputs or training a linear classifier based on embeddings from several layers. We leave this multi-layer approach for future work. To do so, we have to profile numerous parameters across all LLM layers for each request.

Using LLama3-8b-instruct as our model and the Alpaca dataset (Taori et al., 2023), our profiling process begins by extracting embedded elements from each layer during the prefilling and decoding phases. We profile embeddings across all 32 layers for each token, using 1,000 prompts from the dataset, and retain the embedding tensors along with the remaining token counts as training data. The shape of the embedding tensor after the prefilling phase and before decoding is $[1, \text{input token size}, 4096]$, while the embedding tensor for each decoding iteration is $[1, 1, 4096]$.

***Predictor architecture.*** To develop our predictor, we train an MLP using the embeddings from each layer and evaluate prediction accuracy. We use a neural network model consisting of a fully connected network with two linear layers. The first layer maps the input embedding to a 512-dimensional space, followed by a ReLU activation. The second layer outputs predictions for the number of output tokens by classifying the predicted length into one of $k = 10$ equal-width bins, rep-

resenting output lengths between 0 and 512 tokens. The $i$-th bin $b_i$ covers the range $\left[\frac{512i}{10}, \frac{512(i+1)}{10}\right)$. For the final bin, $b_{10}$, it includes the upper boundary and covers the range $[460.8, 512]$.

The model is trained over 30 epochs with a batch size of 32, using the AdamW optimizer to control for overfitting. We employ a cosine annealing schedule to reduce the learning rate from 0.01 to 0 gradually. The loss function is CrossEntropyLoss, appropriate for our multi-class classification task. Here, we present one example of a predictor; our approach can, therefore, be used with any suitable and efficient learning scheme that yields a predictor. For the first token prediction, we compute the average of all token embeddings in the prompt, and this averaged embedding serves as input to the predictor. Preliminary evaluations across all 32 layers (Figure 2) indicate that layers 10-15 provide the most accurate length predictions.

***Focused profiling***. Based on this result, we concentrate our profiling on these layers, gathering over 7 million training pairs. Subsequently, we train the predictor, employing smoothing techniques to improve the accuracy of our length predictions further.

***Smoothing.*** We observe that while the predicted probability vector in each iteration is reasonable, there can be large variance between iterations. We, therefore, employ Bayesian inference (Särkkä & Svensson, 2023) to update the probability estimate as new predictions become available, maintaining a more accurate estimate of the probability distribution over time. In the context of Bayesian inference, the estimate from previous iterations provides a starting point for the current prediction. After observing the current prediction, the initial estimate is updated to reflect the new information. This update is done by adjusting the previous estimate based on how well it aligns with the current prediction and then normalizing it. The updated estimate from this iteration will then be used as the starting point for the next one.

At each iteration $t$, the prior is updated because the predicted length may shift between bins over time. Specifically, let $T \in [0,1]^{k \times k}$ represent the transition matrix, where $T_{i,j}$ denotes the probability that a value in bin $B_j$ at iteration $t-1$ moves to bin $B_i$ at iteration $t$. In our case, transitions can only occur between neighboring bins, specifically from $B_{i+1}$ to $B_i$. We assume that lengths within each bin are uniformly distributed. Therefore, the diagonal entry $T_{i,i}$ reflects the probability that the value remains in the same bin, which is $1 - \frac{1}{\text{bin size}}$, while the off-diagonal entry $T_{i,i+1}$ represents the probability of moving from $B_{i+1}$ to $B_i$, which is $\frac{1}{\text{bin size}}$. $T$ is calculated once from bin sizes, with its structure detailed in Appendix A. The estimated probability $\hat{q}^{(t)}$ at iteration $t$ is computed as follows:

1. Initialize $\hat{q}_{\text{prior}}^{(0)} = p^{(0)}$.

2. At each iteration $t$, update the prior: $\hat{q}_{\text{prior}}^{(t)} = T \cdot \hat{q}_{\text{prior}}^{(t-1)}$.

3. Update the current probability: $\hat{q}^{(t)}(i) = \frac{\hat{q}_{\text{prior}}^{(t)}(i) p^{(t)}(i)}{\sum_j \hat{q}_{\text{prior}}^{(t)}(j) p^{(t)}(j)}$.

The predicted length at iteration $t$ is $L_t = \sum_i \hat{q}^{(t)}(i) \cdot m_i$, where $m_i$ is the average length in bin $B_i$, namely $m_i = \frac{b_i + b_{i+1}}{2} = \frac{128(2i+1)}{5}$. This iterative process refines the prior distribution over time. Figure 3 shows the mean absolute error for predictions from different model layers.

## 3.2 COMPUTING OUTPUT LENGTH PREDICTION PER ITERATION

There are two approaches to computing predictions based on the embedding extracted from intermediate layer(s) (in our case, layer 11) of the LLM. In the first approach, we can compute the prediction directly on the GPU using the embedding available at the selected layer. While this method avoids data transfer, it introduces a slowdown in the decoding phase, as both the prediction computation and the model's forward pass share the same GPU. In the second approach, we can compute the prediction on the CPU. For this, we extract the embedding from layer 11 and transfer it to the CPU. The prediction is then computed in parallel while the LLM processes the remaining layers (in our case, layers 12-32) on the GPU.

We note that whether we implement the length prediction on the GPU or the CPU, the overhead in terms of computational cost is minimal. Specifically, the size of the MLP that we train for the length prediction has around 2.1 *million* parameters, while the overall Llama model we use has 8 *billion*

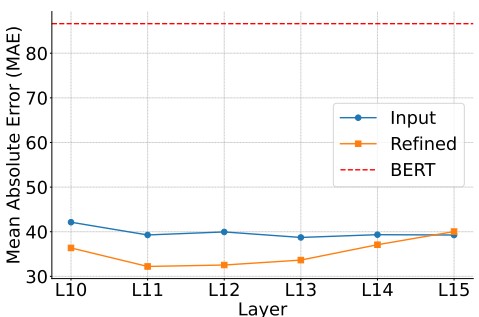

Figure 3: Mean Absolute Error for the predicted length, comparing BERT input embedding (dashed red), average token embedding without refinement (blue), and with refinement (orange), for different layers.

| Device | Batch | Mean ($\mu s$) | Std ($\mu s$) |
|--------|-------|------------|-----------|
| CPU | 1 | 0.155 | 0.0043 |
| CPU | 512 | 9.43 | 3.75 |
| CPU | 1024 | 6.19 | 1.46 |
| CPU | 2048 | 5.94 | 1.09 |
| CUDA | 1 | 0.087 | 0.00017 |
| CUDA | 512 | 0.615 | 0.093 |
| CUDA | 1024 | 0.497 | 0.078 |
| CUDA | 2048 | 0.429 | 0.084 |
| LLM Forward Pass | | | |
| Mean | | 101.10 ms | |

Table 1: Mean and standard deviation of inference time per sample (TPS) for CPU and CUDA with different batch sizes, in microseconds ($\mu s$). Additionally, the LLM forward pass latency per token for Meta-LLaMA-3-8B-Instruct is provided.

parameters. Since the computation (number of FLOPs) per token is roughly scaled with the number of parameters, the overhead of the length prediction is around $0.03\%$. This overhead becomes even more negligible for larger model sizes or when using a smaller MLP for length prediction. Table 1 shows the mean inference time for length prediction of the two approaches.

## 3.3 SCHEDULING POLICY

As background, traditional scheduling typically assumes that request completion times are either completely unknown, in which case FCFS is a natural strategy, or completely known in advance, allowing strategies such as shortest job[1] first (SJF) to minimize average wait time. In many practical systems, exact request completion times are unknown. Several studies have explored queueing systems with predicted, rather than exact, service times, generally to minimize the average time a request spends in the system. Building on previous works, (Mitzenmacher, 2019; Shahout & Mitzenmacher, 2024) analyzed the shortest predicted job first (SPJF) and the shortest predicted remaining processing time (SPRPT) policies in the context of a single server where only one request executes at a time. SPRPT tracks the predicted remaining processing time for each request, allowing preemption if a new request arrives with a shorter predicted remaining service time.

Such previous work, while motivating using predictions, does not take into account the challenge in LLM systems that preemptions introduce memory overhead, particularly in managing the key value (KV) cache (as explained in Section 2). While preemptions can reduce response time, they increase memory consumption, which may ultimately degrade performance due to memory constraints. This is because, in LLM systems, when memory is full, we either discard requests from memory and recompute them once memory is available, or we swap KV cache entries from the GPU to the CPU. Both approaches impact response time, as discarding requires recomputation, and swapping interrupts the forward-pass of the model, causing delays for the entire running batch. Intuitively, we should limit preemptions to avoid exceeding memory constraints; in particular, we recognize that preemptions made earlier in a request's execution consume fewer resources, while preemptions closer to completion should be avoided.

In TRAIL, given the initial prediction $r$ for a request (which we treat as a number corresponding to the middle of its predicted bin), we only allow preemption for the first $\lfloor c \cdot r \rfloor$ iterations for a fixed constant $c$. This restricts preemption as a request nears completion and takes substantial memory.

Interestingly, we can analyze a corresponding theoretical model for single-server SPRPT with limited preemptions, which we believe is interesting in its own right. While this model does not capture the complexity of LLM systems (as it has no notion of memory, and is single-server), we think it also provides insight into the potential of limited preemption. Using standard queueing theory as-

---
[1]The terms job and request are used interchangeably.

sumptions (M/G/1 queuing system with Poisson arrivals of rate $\lambda < 1$ and independent service and prediction times), we can derive a closed-form expression for the mean response time (Lemma 1).

Specifically, the processing times for each arriving request are independent and drawn based on the cumulative distribution $F(x)$ with an associated density function $f(x)$. Predictions, independent over requests, follow the density function $g(x, r)$, where $x$ is the actual size and $r$ is the predicted size. Thus, $\int_{r=0}^{\infty} g(x, r)dr = f(x)$. A request is described by a triple $(x, r, a)$, where $x$ is the actual size, $r$ is the predicted size, and $a$ is request age (time spent serving the request). We set threshold $a_0 = c \cdot r$, with $c$ as a tunable parameter. The system allows preemption when $a < a_0$ and disables it when $a \geq a_0$, balancing early resource use with timely completion. When $c = 1$, the system becomes the same as SPRPT.

The proof of Lemma 1 below is presented in Appendix C. Simulations showing that SPRPT with limited preemption improves memory usage compared to traditional SPRPT are presented in Appendix D. Additionally, in Appendix E, we provide another theoretical model in the same framework for the setting of refined predictions, that may also be of independent interest.

**Lemma 1.** *For SPRPT with limited preemption, where at age $a_0$ the requests become non-preemptable, the expected mean response time for a request of true size $x$ and predicted size $r$ is*

$$\mathbb{E}[T(x,r)] = \frac{\lambda \left( \int_{y=0}^{r} \int_{x_I=0}^{\infty} x_I^2 \cdot g(x_I, y)dx_I dy + \int_{t=r+a_0}^{\infty} \int_{x_I=t-r}^{\infty} g(x_I, t) \cdot (x_I - (t-r))^2 \cdot dx_I dt \right)}{2(1-\rho_r')^2}$$

$$+ \int_0^{a_0} \frac{1}{1 - \rho'_{(r-a)^+}} \, da + (x - a_0).$$

*where $\rho_r' = \lambda \int_{y=0}^{r} \int_{x_I=0}^{\infty} x_I \cdot g(x_I, y)dx_I dy$.*

## 4 EVALUATION

**Setup and implementation.** Our implementation is based on the open-source vLLM system (v0.5.0), with chunked prefill enabled in both our scheduler and the baseline scheduling methods. We set the out-of-memory mode to discard jobs and recompute them once memory becomes available. The evaluation is conducted on a server with a single NVIDIA A100 80GB GPU and 64 AMD EPYC 7313 16-Core Processor cores, with 503 GiB of memory and running CUDA version 12.3. We use *LLama3-8b-instruct* as a serving model with single-GPU. The workload is generated using the Alpaca dataset (Taori et al., 2023), derived from open-source conversational exchanges and originally used to fine-tune the Alpaca model. We sample 10k unique prompts from the dataset for model serving, distinct from those used to train the length predictor.

**Benchmark and Metrics.** The benchmark we use simulates a chatbot serving in a client-server setup, where the server hosts an LLM model, and the client sends requests at a specified request rate. The server runs the vLLM OpenAI API, while the client sends prompts from the Alpaca dataset to mimic a real-world scenario. The benchmark measures the mean and median latency and Time To First Token (TTFT), providing insights into the LLM's responsiveness under varying request loads. In addition to regular load, we evaluate performance under burst conditions.

**Baselines.** We compare our approach with four baselines: (1) vanilla vLLM (Kwon et al., 2023), using First Come First Served; (2) vLLM with Shortest Job First based on BERT predictions and chunked prefill; (3) TRAIL with refined embedding predictions; and (4) TRAIL with BERT predictions.

### 4.1 PREDICTIONS ACCURACY

Figure 3 compares the mean absolute error (MAE) of the BERT predictions from the prompts, our linear classifier predictions from the embedding layers, and the refined predictions after using Bayesian inference on the different layers choices. We further evaluated the accuracy of this refined prediction from layer 11 against the ground truth and compared it to BERT predictions. A heatmap was used to compare the ground truth remaining length with the predicted remaining length across multiple sequences. Both the x-axis (ground truth) and y-axis (predictions) were divided into ten bins, representing the remaining lengths as the sequence progresses: Where the $i$-th bin $b_i$ covers

the range $\left[\frac{512i}{10}, \frac{512(i+1)}{10}\right)$. Each request is counted multiple times in the heatmap, once for each prediction made during its execution. Each cell in the heatmap contains the logarithm of the number of occurrences for a given pair of ground truth and predicted length at different stages. For BERT, where only one prediction is made, we subtract one from the initial predicted length after each generated token to create a comparable heatmap. As shown in Figure 4, the refined predictions from layer-11 embedding are more accurate than the BERT predictions, as they exhibit higher values along the diagonal and lower values off the diagonal (indicating greater prediction accuracy). Meanwhile, the BERT predictions show lower values along the diagonal and higher values off the diagonal, reflecting larger differences between predicted and actual lengths.

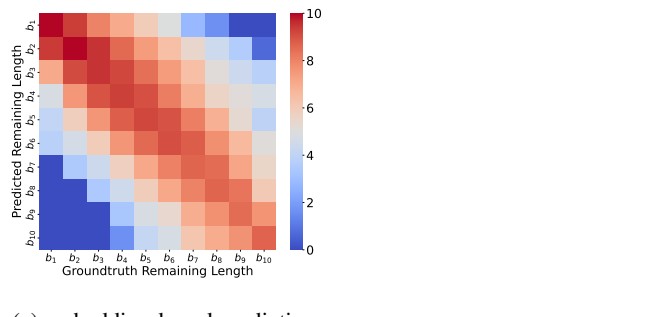

(a) embedding-based predictions

(b) BERT predictions

Figure 4: Log-scaled comparison of ground-truth vs predicted lengths bins. The $i$-th bin $b_i$ covers the range $\left[\frac{512i}{10}, \frac{512(i+1)}{10}\right)$.

## 4.2 LLM SERVING

We begin by comparing the mean latency and TTFT of TRAIL across different values of $c$ ($c = 0.5, 0.8, 1$) while setting the request rate to 14. ($c = 1$ mimics SRPT without limiting preemption.) Figure 5 shows that when $c = 1$, TRAIL exhibits higher mean latency and TTFT, while $c = 0.8$ results in the lowest values, closely followed by $c = 0.5$. We also evaluated the system with $c = 0.2$, which produced higher latency and TTFT compared to the larger $c$ values (though not shown here to focus more closely on this range of $c$ values). These results highlight that while preemption in scheduling benefits LLM systems, it should be limited due to the additional memory overhead it introduces, which can affect serving efficiency. Based on this, we set the default $c$ value for TRAIL to 0.8 in the remainder of the evaluation. The optimal $c$, however, may vary depending on system memory and workload, as it affects the trade-off between preemption benefits and memory overhead.

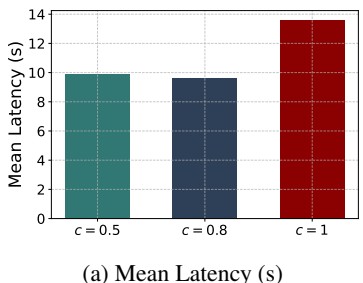 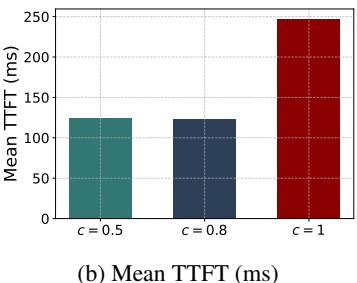

(a) Mean Latency (s)

(b) Mean TTFT (ms)

Figure 5: Comparison of mean latency and TTFT across different values of $c$ ($c = 0.5, 0.8, 1$) at a request rate of 14.

Figure 6 presents the mean and median latency, along with TTFT, as a function of the request rate for four LLM serving systems: (1) vanilla vLLM (Kwon et al., 2023), which uses First Come First Served (FCFS) as the scheduling policy (vLLM-FCFS); (2) vLLM with a Shortest Job First (SJF) scheduling policy for newly scheduled sequences based on BERT predictions from sequence prompts (vLLM-SJF_BERT), which also prioritizes new sequences and implements chunked prefill;

(3) TRAIL with $c = 0.8$ using refined predictions from embeddings with chunked prefill enabled; (4) TRAIL with $c = 0.8$ using BERT predictions (TRAIL-BERT); and (5)TRAIL+ oracle baseline where requests are scheduled based on their exact length. The comparison between TRAIL-BERT and TRAIL evaluates the effectiveness of embedding-based predictions, while the comparison between vLLM (with both scheduling policies) and TRAIL (with both prediction methods) evaluates the impact of limited preemption in LLM systems. The figure shows that SJF based on BERT predictions provides minimal benefits over vLLM-FCFS, as vLLM implementation prioritizes incoming requests over existing running requests. Additionally, both TRAIL and TRAIL-BERT, which implement limited preemption, achieve lower mean and median latency (Figures 11a,11b) and lower mean and median TTFT (Figures 6b,6d) compared to vLLM, with TRAIL using refined embedding predictions showing the lowest latency and TTFT due to more accurate predictions (Figures 3,4).

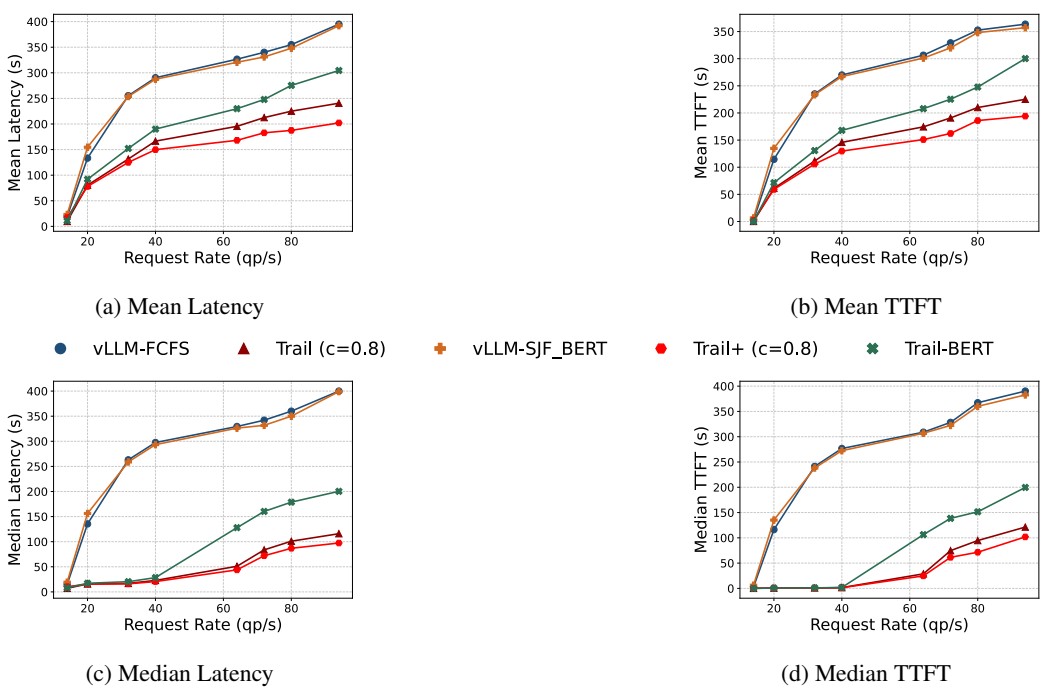

(a) Mean Latency

(b) Mean TTFT

(c) Median Latency

(d) Median TTFT

Figure 6: Mean and median latency, along with TTFT, as a function of request rate for four LLM systems: (1) vLLM-FCFS, vLLM using FCFS; (2) vLLM-SJF_BERT, vLLM using SJF based on BERT; (3) TRAIL with $c = 0.8$ and refined embedding predictions; and (4) TRAIL-BERT with $c = 0.8$ using BERT predictions.

Figure 7 shows the mean and median latency and TTFT for vLLM, vLLM-SJF_BERT, and TRAIL with two different $c$ values ($c = 0.8$ and $c = 1$) under a burst scenario. In this scenario, all requests arrive within a very short time interval at the beginning of the experiment, simulating a sudden spike in demand. The results show that TRAIL continues to offer benefits with lower latency and TTFT, as our implementation ranks all requests (running and waiting) based on length predictions and prioritizes them accordingly, while vLLM prioritizes new requests over existing running ones. However, since no new requests arrive during processing, preemption has no advantage, leading to similar performance between TRAIL with $c = 0.8$ and $c = 1$.

## 5    RELATED WORKS

Recent efforts to optimize the serving of LLMs have been explored. ORCA (Yu et al., 2022) introduced token-level scheduling, assuming that batches are processed at the token level rather than the sequence level. vLLM (Kwon et al., 2023) applies PagedAttention to reduce memory overhead. However, these works rely on first-come, first-served scheduling, which could face head-of-line blocking.

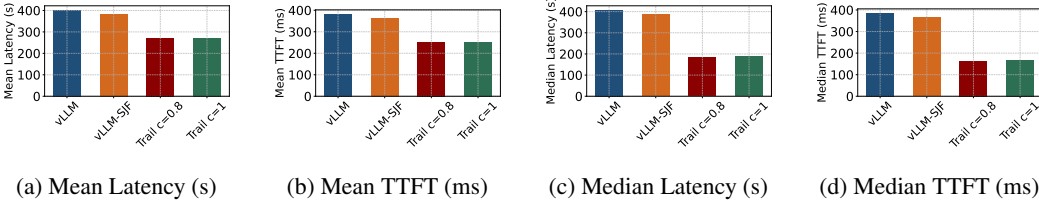

| (a) Mean Latency (s) | (b) Mean TTFT (ms) | (c) Median Latency (s) | (d) Median TTFT (ms) |

Figure 7: Mean, Median of Latency and TTFT when we have burst of requests

In traditional scheduling, methods like shortest job first (SJF) and shortest remaining processing time (SRPT) aim to minimize response times, while multi-level feedback queues (MLFQ) adjust job priorities dynamically without prior knowledge of job sizes. FastServe (Wu et al., 2023) uses MLFQ to avoid head-of-line blocking. However, this leads to frequent preemptions, which increase the cost of managing KV cache memory and offloading to the CPU. Several works have proposed prediction-based scheduling to address these challenges. Zhen et al. (Zheng et al., 2024) enhanced LLM inference by predicting response lengths with an additional LLM model and scheduling them according to length predictions. While this approach optimizes inference, it introduces prediction overhead. Other works predict output lengths using machine learning models such as DistilBERT and OPT, improving resource allocation and reducing memory issues during inference. The works in (Jin et al., 2023; Stojkovic et al., 2024; Cheng et al., 2024) approach length prediction as a classification problem, while other works (Qiu et al., 2024b;a) adopt regression-based techniques.

LTR (Fu et al., 2024) employs a Learning-to-Rank approach similar to TRAIL. Instead of predicting the absolute output size of a request, LTR ranks requests based on their output size, allowing the system to prioritize those with fewer remaining tokens. Although ranking is a simpler task than absolute size prediction, it requires training a ranking model in an offline phase. A limitation of LTR is that it ignores the size of the prompt when ranking, considering only the output size.

## 6 LIMITATIONS AND CONCLUSION

We have presented TRAIL, a novel approach that significantly improves response time in LLM inference. Our method is built on two key contributions. First, we obtained low-overhead, high-accuracy predictions from the target LLM itself. After generating each output token, we recycle the embedding of its internal structure as input for a lightweight classifier. This classifier predicts the remaining length for each running request. We demonstrated the high accuracy of this approach and its low overhead compared to existing prediction methods that rely solely on sequence prompts. Specifically, our refined predictions from layer embeddings achieve 2.66x lower mean absolute error compared to BERT predictions from sequence prompts. Second, utilizing these predictions for output lengths, we proposed a Shortest Remaining Processing Time (SRPT) variant with limited preemption. This variant is specifically designed to account for memory overhead in LLM systems. We derived a closed-form formula for this SRPT variant in an M/G/1 queue model and simulated it to demonstrate its potential. By integrating TRAIL with vLLM, we have demonstrated improvements in overall latency and time-to-first-token metrics using real-world datasets. Our experimental results show that TRAIL achieves 1.66x to 2.01x lower mean latency and 1.76x to 24.07x lower mean time to first token compared to vLLM tested using the Alpaca dataset.

**Limitations.** One key challenge is to select the most informative layer embedding to use as input for the linear classifier. Due to time and resource constraints, we focused only on the Alpaca dataset, as profiling embeddings across all layers is computationally demanding. There are several promising directions for future research. We plan to explore the relationship between layer selection and prediction accuracy across multiple datasets. Another direction is leveraging multiple-layer embeddings through weighted averaging to enhance prediction accuracy. Additionally, experimenting with logarithmic bin sizes for the linear classifier could offer further benefits. A potential optimization is to compute embedding predictions at specific intervals rather than every iteration, reducing computational overhead. Lastly, to address data drifts and maintain performance over time, we aim to explore the dynamic retraining of the linear classifier.

## ACKNOWLEDGEMENTS

We thank Sham Kakade for providing access to the Kempner cluster for evaluation. We also thank Yonatan Belinkov for the helpful discussions. Rana Shahout was supported in part by the Schmidt Futures Initiative and Zuckerman Institute. Michael Mitzenmacher was supported in part by NSF grants CCF-2101140 and DMS-2023528. Rana Shahout, Minlan Yu, and Michael Mitzenmacher are partially supported by NSF CNS NeTS 2107078. This work was supported in part by ACE, one of the seven centers in JUMP 2.0, a Semiconductor Research Corporation (SRC) program sponsored by DARPA. We are grateful for support from the DARPA ASKEM project (Award HR00112220042) and the ARPA-H Biomedical Data Fabric project.

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

# Appendix

## Table of Contents

## A BAYESIAN INFERENCE TRANSITION MATRIX

The transition matrix captures transitions between neighboring bins based on the assumption of uniform distribution within each bin. The diagonal and sub-diagonal entries reflect the probabilities of remaining in the same bin or moving to the adjacent lower bin, respectively. Below is the explicit form of the transition matrix for a general number of bins.

$$
T = \begin{bmatrix}
1 - \frac{1}{\text{bin size}} & 0 & 0 & \dots & 0 \\
\frac{1}{\text{bin size}} & 1 - \frac{1}{\text{bin size}} & 0 & \dots & 0 \\
0 & \frac{1}{\text{bin size}} & 1 - \frac{1}{\text{bin size}} & \dots & 0 \\
\vdots & \vdots & \vdots & \ddots & \vdots \\
0 & 0 & 0 & \dots & 1 - \frac{1}{\text{bin size}}
\end{bmatrix}
$$

## B SOAP ANALYSIS

We utilize the SOAP framework (Scully & Harchol-Balter, 2018) to derive precise expressions for the mean response time. SOAP also provides the Laplace-Stieltjes transform of the response time distribution, but we focus on mean response time for ease of comparison throughout the paper.

The SOAP framework is designed to analyze scheduling policies for M/G/1 queues that can be characterized by rank functions. Recent work by Scully and Harchol-Balter (Scully & Harchol-Balter, 2018) has categorized a broad range of scheduling policies as belonging to the SOAP class. These policies determine job scheduling based on ranks, always prioritizing the job with the lowest rank. (If multiple jobs share the same rank, First Come First Served is used as the tiebreaker.) The rank function itself is a key component, assigned to each job based on certain static properties, typically referred to as the job's type or descriptor. For instance, the descriptor could indicate the job's class in multi-class models, as well as static characteristics like its service time (job size). The rank may also depend on the job's age, or the time it has already been served.

SOAP policies assume that a job's rank is influenced only by its inherent characteristics and its age, aligning well with the model and scheduling algorithm used in TRAIL. For more detailed information, the reader is referred to (Scully & Harchol-Balter, 2018).

In SOAP analysis, the tagged-job technique is employed. Specifically, we track a tagged job $J$ with size $x_J$ and descriptor $d_J$, and use $a_J$ to represent the time $J$ has already been served. The mean response time for $J$ is calculated as the sum of its waiting time (the period from its arrival to the start of service) and its residence time (the duration from the start of service to its completion). To calculate the waiting time, SOAP evaluates the delays caused by both old jobs that arrived before $J$ and new jobs that arrive after $J$. A key concept in this analysis is the worst future rank of a job, as job ranks may change over time. The worst future rank for a job with descriptor $d_J$ and age $a_J$ is denoted by $rank_{d_J}^{\text{worst}}(a_J)$. When $a_J = 0$, the rank function is represented as $r_{worst} = rank_{d_J}^{\text{worst}}(0)$.

In the SOAP framework, the waiting time is equivalent to the transformed busy period in an M/G/1 queue, where the arrival rate is $\lambda$ and the job size is described by $X^{\text{new}}[rank_{d_J}^{\text{worst}}(a)]$. Where $X^{\text{new}}[rank_{d_J}^{\text{worst}}(a)]$ is a random variable representing the time a newly arrived job will be served until it either completes or its rank exceeds $rank_{d_J}^{\text{worst}}(a)$. The initial workload during this period corresponds to the delays caused by old jobs. To account for the delays from old jobs, SOAP transforms the system by categorizing jobs according to their rank. Old jobs that exceed the rank threshold $r_{worst}$ are classified as *discarded* and are not included in the transformed system. Those with ranks at or below $r_{worst}$, referred to as *original* jobs, are treated as arriving at rate $\lambda$ with a specific size distribution $X_0^{\text{old}}[r_{worst}]$. Where $X_0^{\text{old}}[r_{worst}]$ is a random variable representing the service time of an original job with respect to rank $r_{worst}$. Recycled jobs, which were once above the threshold but have now fallen below, are treated as server vacations of length $X_i^{\text{old}}[r_{worst}]$. Where $X_i^{\text{old}}[r_{worst}]$ is a random variable representing the service time of a recycled job for the $i$-th time with respect to rank $r_{worst}$ for $i \geq 1$. In TRAIL, jobs are recycled only once, so we only consider $X_1^{\text{old}}[r_{worst}]$.

SOAP shows that, due to Poisson arrivals see time averages, the stationary delay caused by old jobs follows the same distribution as the queuing time in the transformed M/G/1/FCFS system, characterized by sparse server vacations where original jobs arrive at rate $\lambda$ and follow the size distribution $X_0^{\text{old}}[r_{worst}]$.

**Theorem 1** (Theorem 5.5 of (Scully & Harchol-Balter, 2018)). *Under any SOAP policy, the mean response time of a job with descriptor $d$ and size $x$ is:*

$$
\mathbb{E}[T(x,d)] = \frac{\lambda \cdot \sum_{i=0}^{\infty} \mathbb{E}[X_i^{old}[r_{worst}]^2]}{2(1 - \lambda\mathbb{E}[X_0^{old}[r_{worst}]])(1 - \lambda\mathbb{E}[X^{new}[r_{worst}]])}
$$
$$
+ \int_0^x \frac{1}{1 - \lambda\mathbb{E}[X^{new}[rank_{d_J}^{worst}(a)]]} da.
$$

## C   SPRPT WITH LIMITED PREEMPTION

We begin by describing the rank function for a job with descriptor $(x, r, a)$, where $x$ is the actual size, $r$ is the predicted size, and $a$ is the job's age. The threshold where we stop preemption from occurring is set as $a_0 = c \cdot r$. The rank function is:

$$
rank(x, r, a) = \begin{cases} r - a & \text{if } a < a_0 \\ -\infty & otherwise \end{cases}
$$

As this rank function is monotonic, A job's worst future rank is its initial prediction:

$$
rank_{d,x}^{\text{worst}}(a) = \begin{cases} r - a & \text{if } a < a_0 \\ -\infty & \text{otherwise} \end{cases} \tag{1}
$$

When $a = 0$, the rank function is denoted by $r_{worst} = rank_{d,x}^{\text{worst}}(0) = r$.

**Lemma 1.** *For SPRPT with limited preemption, where at age $a_0$ the requests become non-preemptable, the expected mean response time for a request of true size $x$ and predicted size $r$*

*is*

$$\mathbb{E}[T(x,r)] = \frac{\lambda \left( \int_{y=0}^{r} \int_{x_I=0}^{\infty} x_I^2 \cdot g(x_I, y) dx_I dy + \int_{t=r+a_0}^{\infty} \int_{x_I=t-r}^{\infty} g(x_I, t) \cdot (x_I - (t - r))^2 \cdot dx_I dt \right)}{2(1 - \rho_r')^2}$$

$$+ \int_0^{a_0} \frac{1}{1 - \rho_{(r-a)^+}'} \, da + (x - a_0).$$

*where $\rho_r' = \lambda \int_{y=0}^{r} \int_{x_I=0}^{\infty} x_I \cdot g(x_I, y) dx_I dy$.*

*Proof.* To analyze SPRPT with limited preemption using SOAP, based on the worst future rank (equation 1), we calculate $X^{\text{new}}[rank_d^{\text{worst}}(a)]$, $X_0^{\text{old}}[r_{worst}]$ and $X_i^{\text{old}}[r_{worst}]$ for a tagged job $J$ with descriptor $(x, r, a)$.

### C.0.1 $X^{\text{NEW}}[rank_{d,x}^{\text{WORST}}(a)]$ COMPUTATION:

Suppose that a new job $K$ of predicted size $r_K$ arrives when $J$ has age $a$. If $a < a_0$ and $K$ has a predicted job size less than $J$'s predicted remaining process time $(r - a)$, $K$ will always outrank $J$. Thus

$$X_{x_K}^{\text{new}}[r - a] = x_K \mathbf{1}(r_K < r - a)\mathbf{1}(a < a_0)$$

$$\mathbb{E}[X^{new}[r - a]] = \int_0^{r-a} \int_{x_K=0}^{\infty} x_K \cdot g(x_K, y) dx_K dy$$

### C.0.2 $X_0^{\text{OLD}}[r_{worst}]$ COMPUTATION:

Let $I$ be an old job in the system. Whether job $I$ is an original or recycled job depends on its predicted size relative to J's predicted size. If $r_I \leq r$, then $I$ is original until its completion because its rank never exceeds $r$.

$$X_{0,x_I}^{\text{old}}[r] = x_I \mathbf{1}(r_I \leq r).$$

$$\mathbb{E}[X_0^{\text{old}}[r]] = \int_{y=0}^{r} \int_{x_I=0}^{\infty} x_I \cdot g(x_I, y) dx_I dy.$$

$$\mathbb{E}[(X_0^{\text{old}}[r])^2] = \int_{y=0}^{r} \int_{x_I=0}^{\infty} x_I^2 \cdot g(x_I, y) dx_I dy.$$

### C.0.3 $X_i^{\text{OLD}}[r_{worst}]$ COMPUTATION:

If $r_I > r$, then $I$ starts discarded but becomes recycled when $r_I - a = r$. This means at age $a = r_I - r$ and served till completion only if $a < a_0$, which will be $x_I - a_I = x_I - (r_I - r)$:

Thus, we have
$$X_{1,x_I}^{\text{old}}[r] = x_I - (r_I - r).$$

For $i \geq 2$,
$$X_{i,x_I}^{\text{old}}[r] = 0.$$

$$\mathbb{E}[X_1^{\text{old}}[r]^2] = \int_{r_I=r+a_0}^{\infty} \int_{x_I=r_I-r}^{\infty} g(x_I, t) \cdot (x_I - (r_I - r))^2 \cdot dx_I dr_I.$$

Applying Theorem 5.5 of SOAP (Scully & Harchol-Balter, 2018) yields that the mean response time of jobs with descriptor $(r)$ and size $x$ is as follows. Let

$$\rho_r' = \lambda \int_{y=0}^{r} \int_{x_I=0}^{\infty} x_I \cdot g(x_I, y) dx_I dy.$$

Then

$$\mathbb{E}[T(x,r)] = \frac{\lambda \left( \int_{y=0}^{r} \int_{x_I=0}^{\infty} x_I^2 \cdot g(x_I,y) dx_I dy + \int_{t=r+a_0}^{\infty} \int_{x_I=t-r}^{\infty} g(x_I,t) \cdot (x_I - (t-r))^2 \cdot dx_I dt \right)}{2(1-\rho_r')^2}$$
$$+ \int_{0}^{a_0} \frac{1}{1 - \rho_{(r-a)^+}'} \, da + (x - a_0).$$

Let $f_p(y) = \int_{x=0}^{\infty} g(x,y) dx$. Then the mean response time for a job with size $x$, and the mean response time of all jobs are given by

$$\mathbb{E}[T(x)] = \int_{y=0}^{\infty} f_p(y) \mathbb{E}[T(x,y)] dy,$$

$$\mathbb{E}[T] = \int_{x=0}^{\infty} \int_{y=0}^{\infty} g(x,y) \mathbb{E}[T(x,y)] dy dx.$$

$\square$

## D    SIMULATION OF SPRPT WITH LIMITED PREEMPTION

In our simulation, memory usage is modeled as proportional to the age of each job: the amount of service it has received so far. This reflects the idea that jobs consume more memory the longer they remain in the system. We measure the response time to evaluate scheduling efficiency along with memory consumption.

We consider a single-queue setting with Poisson arrivals, where the job service time follows an exponential distribution with mean 1 ($f(x) = e^{-x}$). Two prediction models are used: 1) Exponential predictions (Mitzenmacher, 2019), where the prediction for a job with service time $x$ is also exponentially distributed with mean $x$, given by $g(x,y) = e^{\frac{-x-y}{x}}$. 2) A "perfect predictor", where the prediction is always accurate, given by $g(x,y) = e^{-x}$.

Figure 8 shows the mean response time (blue line) and the peak memory usage (green line) under various arrival rates and with different $c$ values. The key takeaway is that limiting preemption; reducing how often jobs are interrupted and rescheduled, can lead to better memory utilization while maintaining a reasonable response time.

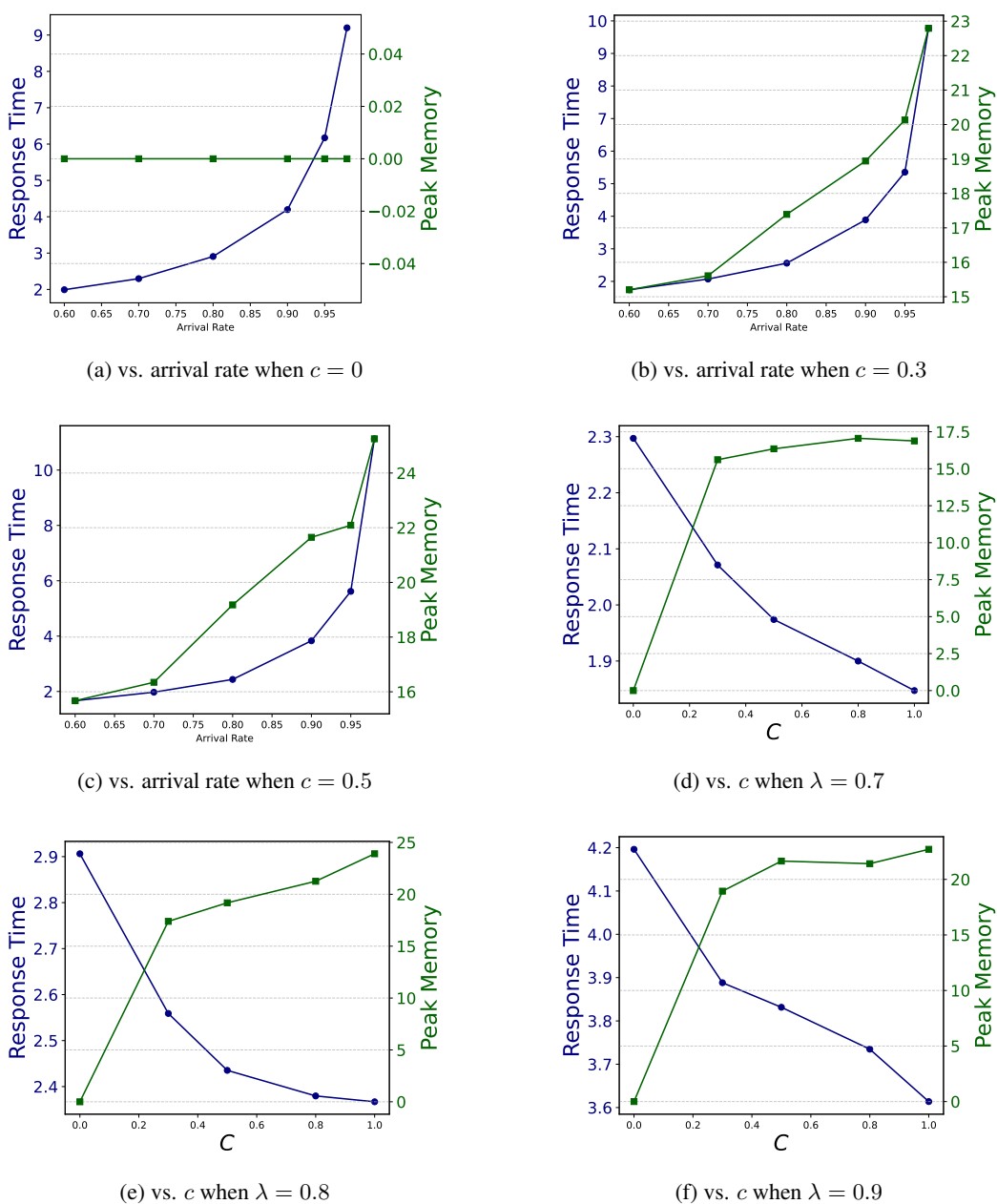

Figure 8: Comparing memory usage and response time across different arrival rates and values of $c$.

The parameter $c$, used to manage the trade-offs in memory utilization, is influenced by the KV cache's available memory, which depends on the model size, batch sizes, and incoming request sizes. In scenarios with a mix of long and short requests, an appropriately chosen $c$ value can prevent memory monopolization by long requests, thereby improving throughput. The value of $c$ can also be dynamically adjusted in real time.

# E    REFINED PREDICTION FRAMEWORK

We consider M/G/1 queueing systems with arrival rate $\lambda$. The processing times for each arriving job are independent and drawn based on the cumulative distribution $F(x)$, with an associated density function $f(x)$.

Given a job of size $X = x$, and a sequence of refined predictions $Y_0, Y_1, \ldots, Y_x$, where $Y_i$ represents the prediction after processing $i$ units of $x$, we model the refined predictions as follows:

The probability of the refined prediction $y_i$ at step $i$, given the actual size $x$ and the entire history of previous predictions, is denoted as:

$$P(Y_i = y_i \mid X = x, Y_{i-1} = y_{i-1}, Y_{i-2} = y_{i-2}, \ldots, Y_0 = y_0)$$

Under the Markovian assumption, the refined prediction in step $i$ depends only on the actual size $x$ and the previous prediction $y_{i-1}$:

$$P(Y_i = y_i \mid X = x, Y_{i-1} = y_{i-1}, Y_{i-2} = y_{i-2}, \ldots, Y_0 = y_0) = P(Y_i = y_i \mid X = x, Y_{i-1} = y_{i-1})$$

The joint probability of the sequence of predictions $Y_0, Y_1, \ldots, Y_x$ can be modeled as:

$$P(Y_0 = y_0, Y_1 = y_1, \ldots, Y_x = y_x \mid X = x) = P(Y_0 = y_0 \mid X = x) \prod_{i=1}^{x} P(Y_i = y_i \mid X = x, Y_{i-1} = y_{i-1})$$

The density function $g(x, y)$ corresponding to the list of predictions $y = [y_0, y_1, \ldots, y_{x-1}]$, where each $y_i$ is obtained after processing one unit of job $x$, is obtained by summing over all possible refined predictions:

$$g(x, y) = \sum_{y_0=0}^{\infty} \sum_{y_1=0}^{\infty} \cdots \sum_{y_{x-1}=0}^{\infty} P(Y_0 = y_0, Y_1 = y_1, \ldots, Y_{x-1} = y_{x-1} \mid X = x)$$

### E.1 SPRPT WITH REFINED PREDICTIONS

The relevant attributes are size and the refined prediction list $r$. We can model the system using descriptor $\mathcal{D} = (\text{size, list of refined predictions, age})$.

Let $J$ be a job with a descriptor $(x, r, a)$, where $x$ is J's size, $r$ is a list of refined predictions for J, each generated after processing one unit of the job, and $a$ is J's age. Thus, SPRPT with refined predictions has a rank function $rank(x, r, a) = r[a] - a$ for job of size $x$ and predicted size $r[a]$ at time $a$. We denote the maximum rank function as $r_{max}$.

$J$'s worst future rank is:

$$rank_{d,x}^{\text{worst}}(a) = \sup_{a < b < x} (r[b] - b) = \max_{a < b < x} (r[b] - b)$$

When $a = 0$, the rank function is denoted by $r_{worst} = rank_{d,x}^{\text{worst}}(0) = r_{max}$.

To analyze SPRPT with refined predictions using SOAP, the calculation involves solving a multidimensional summation. While obtaining a closed-form expression for such sums may be challenging, in this section, we propose a method to compute the relevant components ($X^{\text{new}}[rank_d^{\text{worst}}(a)]$, $X_0^{\text{old}}[r_{worst}]$ and $X_i^{\text{old}}[r_{worst}]$) for the SOAP analysis without requiring a closed-form solution.

Let $J$ be a tagged job with descriptor $(x, r, a)$, based on its worst future rank.

$X^{\text{new}}[rank_{d,x}^{\text{worst}}(a)]$ **computation:** Suppose that a new job $K$ of predicted size $r_K$ arrives when $J$ has age $a$. $K$ has a lower rank than the worst possible rank of job $J$ is given by:

$$X_{x_K}^{new}[rank_{d,x}^{\text{worst}}(a)] = \sum_{a_K=0}^{x_K} \mathbf{1}\left(r_K[a_K] - a_K < \max_{a<b<x}(r[b] - b)\right) \cdot \mathbf{1}\left(\forall j < a_I, \, r_I[j] - j < \max_{a<b<x}(r[b] - b)\right)$$

where $\mathbf{1}(\cdot)$ is the indicator function, which equals 1 if the condition inside is true and 0 otherwise.

$$\mathbb{E}\left[X^{\text{new}}\left[\text{rank}_{d,x}^{\text{worst}}(a)\right]\right] = \sum_{x_K=0}^{\infty}\left(\sum_{a_K=0}^{x_K}\mathbf{1}\left(r_K[a_K]-a_K<\max_{a<b<x}(r[b]-b)\right)\right.$$

$$\cdot\mathbf{1}\left(\forall j<a_I,\ r_I[j]-j<\max_{a<b<x}(r[b]-b)\right)$$

$$\left.\cdot g(x_K,y)\right)$$

where the density function $g(x_K, y)$ is defined as:

$$g(x_K, y) = \sum_{y_0=0}^{\infty}\sum_{y_1=0}^{\infty}\cdots\sum_{y_{x_K-1}=0}^{\infty}P(Y_0=y_0, Y_1=y_1,\ldots,Y_{x_K-1}=y_{x_K-1}\mid X=x_K)$$

$X_0^{\text{old}}[r_{worst}]$ **computation:** Let $I$ be an old job in the system. Whether job $I$ is an original depends on its predicted size relative to J's predicted size. $I$ is original till it is exceeds $r_{max}$. we terminate the summing once you encounter an age where $r_I[a_I] - a_I > r_{max}$:

$$X_{0,x_I}^{\text{old}}[r_{max}] = \sum_{a_I=0}^{x_I}\mathbf{1}\left(r_I[a_I]-a_I<r_{max}\right)\cdot\mathbf{1}\left(\forall j<a_I,\ r_I[j]-j<r_{max}\right)$$

$X_i^{\text{old}}[r_{worst}]$ **computation:** If $r_I > r$, then $I$ starts discarded but becomes recycled when $r_I[a]-a = r_{worst}$. This means at age $a = r_I[a_I] - r_{worst}$ and served till it will be again about $r_{worst}$,

Let $i$ be an interval index. The $i$-interval for job $I$ is defined as the interval of ages during which the rank of job $I$ is less than $r_{max}$. Specifically:

$$b_0[r] = 0$$

This indicates that the first interval starts at age 0.

$$c_0[r] = \inf\{a>0\mid r_I(a)-a>r_{max}\}$$

Here, $c_0[r]$ is the smallest age greater than 0 where the rank function $r_I(a)-a$ becomes greater than $r_{max}$, marking the end of the first interval where the rank is less than $r_{max}$.

For subsequent intervals $i \geq 1$:

$$b_i[r] = \inf\{a\geq c_{i-1}[r]\mid r_I(a)-a<r_{max}\}$$

This is the starting point of the $i$-th interval, which begins as soon as the rank $r_I(a)-a$ drops below $r_{max}$ again after the previous interval $[b_{i-1}[r], c_{i-1}[r]]$.

$$c_i[r] = \inf\{a>b_i[r]\mid r_I(a)-a>r_{max}\}$$

Here, $c_i[r]$ is the endpoint of the $i$-th interval, where the rank $r_I(a)-a$ becomes greater than $r_{max}$ again.

For $i \geq 0$, the $i$-interval work is a random variable, written $X_i^{\text{OLD}}[r]$, representing the sum of the tokens where job $I$ has rank less than $r_{max}$ within its $i$-interval.

$$X_i^{\text{OLD}}[r] = \begin{cases} 0 & \text{if } X_d < b_i[r] \\ X_d - b_i[r] & \text{if } b_i[r] \leq X_d < c_i[r] \\ c_i[r] - b_i[r] & \text{if } c_i[r] \leq X_d \end{cases}$$

If $b_i[r] = c_i[r] = \infty$, we define $X_i^{\text{OLD}}[r] = 0$.

## F    DATASET ANALYSIS

The first figure illustrates the cumulative distribution function (CDF) of the Alpaca (Taori et al., 2023) dataset's output lengths, showcasing the variation in output sizes.

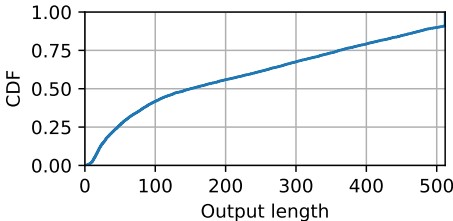

Figure 9: Cumulative Distribution Function (CDF) of output lengths in the dataset.

The second figure presents a histogram of the dataset binned into uniform-sized bins as described in Section 3.1, highlighting the number of prompts within each bin.

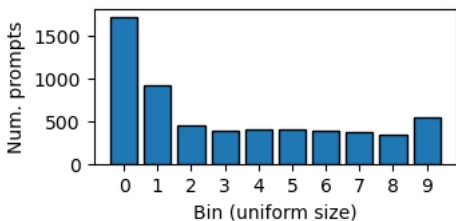

Figure 10: Histogram of the dataset binned into uniform-sized bins.

## G    MULTI-GPU EVALUATION

**Testbed.** For testing multi-GPU settings, we used a machine with dual AMD EPYC 7313 CPUs (16 cores per CPU, totaling 64 threads), 503 GB of RAM, and two NVIDIA A100 GPUs with 80 GB memory each connected via NVLink.

For the first experiment, we used the same serving model (LLama3-8b-instruct), distributed across two GPUs using tensor parallelism. As shown in Figure 11, TRAIL continues to perform effectively.

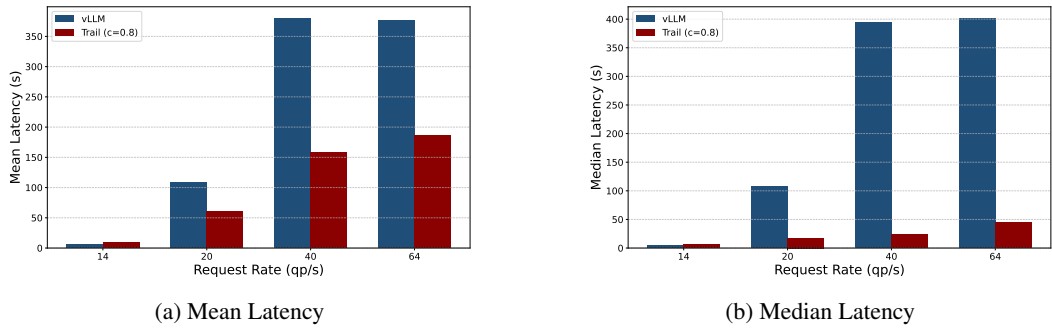

(a) Mean Latency                              (b) Median Latency

Figure 11: Mean and median of end-to-end latency with LLama3-8b-instruct as a serving model distributed across two GPUs using tensor parallelism.

For the second experiment, we tested TRAIL+ (TRAIL with perfect prediction) using Vicuna 13B, distributed across two GPUs using tensor parallelism.

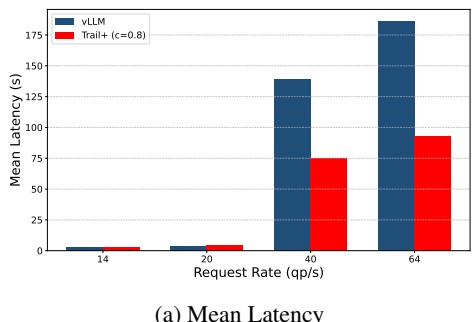

(a) Mean Latency

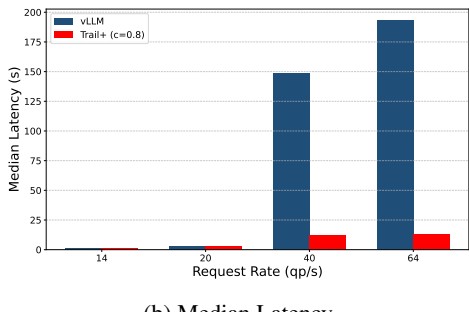

(b) Median Latency

Figure 12: Mean and median end-to-end latency using Vicuna 13B model, distributed across two GPUs using tensor parallelism. Tested with sampling 10000 prompts the first 1000 prompts of the Alpaca dataset.

## H  MINISTRAL-3B-INSTRUCT PREDICTIONS

We evaluated the predictive performance of another LLM, Ministral-3b-instruct, using 1,000 prompts from the Alpaca dataset. The dataset was split such that 75% of the prompts were used for training and the remaining 25% for evaluation.

These results are preliminary. Figure 13 illustrates the Mean Absolute Error (MAE) for predictions across different layers of the model, while Figure 3 demonstrates the refined predictions for each layer after applying the smoothing method described in Section 3.1. From the results, we observe that a specific layer, Layer 10, consistently yields the most accurate predictions.

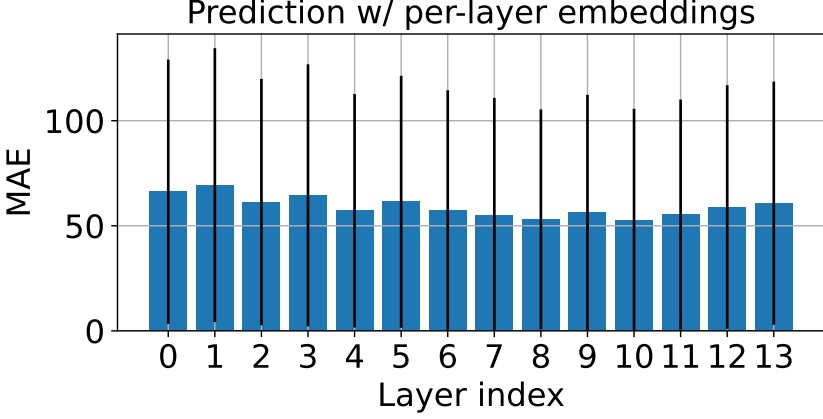

Figure 13: Mean Absolute Error (MAE) per layer for Ministral-3b-instruct predictions.

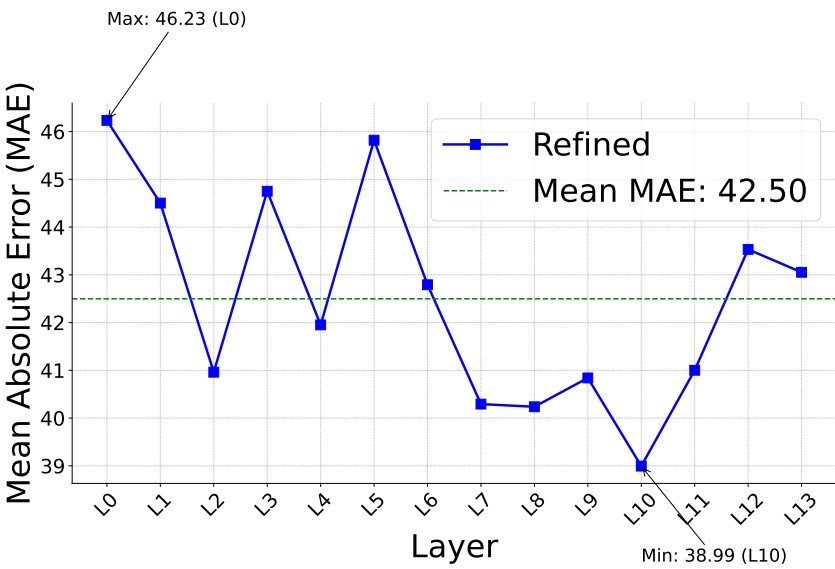

Figure 14: Refined predictions per layer using the smoothing method described in Section 3.1.

