# OpenReview forum: "DON’T STOP ME NOW: EMBEDDING BASED SCHEDULING FOR LLMS"
_ICLR.cc/2025/Conference — ICLR 2025 Poster_

### Official Review · Reviewer_bvWq · 2024-11-02

**Soundness:** 3
**Presentation:** 3
**Contribution:** 2
**Rating:** 5
**Confidence:** 3

**Summary:**

The authors address the scheduling of LLM inference jobs. In particular, to provide a high-quality experience, LLM inference should be fast enough to allow real-time conversational user interaction. Scheduling jobs according to their length has since long been shown to minimize average waiting time, but it requires knowing the length of jobs in advance. Due to its autoregressive nature, LLM-based text generation can lead to highly varying execution times. To overcome this limit, related work proposes various models to predict the length of LLM inference. The authors here improve on this by, rather than using smaller LLM models such as BERT-like ones, predicting the inference length based on the state of the internal layers of the LLM. Moreover, the authors also address the drawback of job preemption, which requires saving intermediate state, which can be considerable for modern LLMs with consequent issues of either exhausting GPU memory or costly memory transfers, by only allowing preemption at the initial steps of a job. In contrast, jobs close to termination can not be preempted. The authors evaluate both the accuracy of their intermediate layer-based job length prediction and preemption-tuned scheduling against three baselines covering vanilla and state-of-the-art solutions.

**Strengths:**

+ Significantly lower mean latency compared to the considered baselines.
+ The model used to predict the LLM output length is way simpler than the ones used by SOTA-related work, which reduces the overall computational burden of LLMs (especially in the context of energy efficiency over the whole sector).
+ Theoretical proof of the proposed scheduling scheme with limited preemption and dynamic threshold to adapt to the variability of inference lengths.

**Weaknesses:**

- Layer-based prediction only tested on one LLM, which makes it unclear how well the method generalizes to other LLMs.
- The choice of the layer used for prediction is model-dependent and requires a preliminary study.
- A deeper sensitivity analysis of the parameter c would have been nice to analyze the optimal value for c better.
- Lack of some design rationales (see questions below)

To give more meaning to the average prediction error presented in Figure 2, it woudl have been nice to provide soem statistics on the inference length.

**Questions:**

The rationale for the number and size of bins for predicting the LLM inference length is missing. What is the mean length of responses? New models have significantly larger contexts. Does this influence the choice? How? Also, would it not make more sense to choose a divider of 512 instead of ten (e.g., 8 or 16) since length is intrinsically an integer?

Learning rate decreased to 0 does not seem to make much sense. I guess it is close to 0. Please check.

Length prediction overhead is small in relative terms but could be significant in absolute terms, especially from an energy point of view. It would be nice to comment on this. Also, how are batches accumulated in this scenario since you want to predict after each next token has been generated? What is the prediction time with a batch size of one?

---

> ### Author Response · Authors · 2024-11-23
>
> Thank you for recognizing the strengths of our work, including the gains in mean latency, the simplicity of our prediction model compared to SOTA methods, and the theoretical proof of our proposed scheduling scheme.
>
> - “Layer-based prediction only tested on one LLM, which makes it unclear how well the method generalizes to other LLMs.”
>
> We agree that generalization is an important concern. Due to resource and time constraints during the rebuttal period, we were unable to evaluate additional LLMs. However, we plan to include experiments with another LLM in the camera-ready version.
>
> - “A deeper sensitivity analysis of the parameter c would have been nice.”
>
> The value of 'c' primarily depends on the available memory for the KV cache, which is influenced by model size, batch sizes, and incoming request sizes. For instance, preempting long requests could monopolize memory and reduce throughput during periods with long requests followed by short ones before finishing. As shown in Appendix D (Figure 8), workload simulations can help determine suitable 'c' values, which can also be adjusted in real time without further system changes. This adjustment directly impacts the rank of jobs (see Equation 1 in Appendix C), connecting the 'c' value to the request ranking.
>
>
> - “The rationale for the number and size of bins for predicting the LLM inference length is missing. What is the mean length of responses? New models have significantly larger contexts. Does this influence the choice?”
>
> We followed the settings of $S^3$[1], which also uses the Alpaca dataset. For the camera-ready version, we will explore non-uniform bin sizes (e.g., bins of geometrically increasing size such as 2, 4, 8, …) to address this.
> In the revised paper, we have added the mean and distribution of response lengths (Appendix F) to provide additional context. For future models with larger contexts, we agree that adjusting bin sizes, including exploring non-uniform bins, may better capture the distribution. We will explore alternatives in the future.
>
> [1] $S^3$: Increasing GPU Utilization during Generative Inference for Higher Throughput, Jin, Yunho, Neurips 2023.
>
> - ”Learning rate decreased to 0 does not seem to make much sense. I guess it is close to 0. Please check.”
>
> We use the minimal learning rate default parameter from the PyTorch implementation of the learning rate decay schedule (https://pytorch.org/docs/stable/generated/torch.optim.lr_scheduler.CosineAnnealingLR.html), which indeed decays the learning rate to zero at the end of training. This is a fairly common choice, which is suggested, for example, in [2].
>
> [2] SGDR: STOCHASTIC GRADIENT DESCENT WITH WARM RESTARTS, Ilya Loshchilov & Frank Hutter, ICLR 2017
>
>
> - “Length prediction overhead is small in relative terms but could be significant in absolute terms, especially from an energy point of view.”
>
> We acknowledge the importance of minimizing prediction overhead, particularly from an energy perspective. As noted in the limitations section of the paper, one potential optimization is to compute embedding predictions at specific intervals rather than every iteration, which could significantly reduce computational costs. Additionally, we have added the LLM forward pass latency per token to Table 1 in the revised paper to clarify the computational overhead associated with predictions in our tested model.

---

> > ### Comment · Reviewer_bvWq · 2024-11-25
> >
> > Commenting on the response points in order:
> >
> > - It is a pity you could not share some preliminary results on another LLM, but I appreciate the promise to add it for the CR version.
> >
> > - I suggest summarising this observation on the parameter c in the main article.
> >
> > - This depends if T_cur goes from 0 to T_max-1 or from 1 to T_max (my guess is the first, but I did not check the code).
> >
> > - Here, I'm still missing the prediction time with batches of size one or the time needed to collect enough predictions to fill a batch of size, e.g., 512, as reported in the paper.
> >
> > Based on the above I would keep my current score.

---

> ### Author Response · Authors · 2024-11-27
>
> Thank you for your thoughtful feedback and for engaging with our responses.
>
> - We appreciate your understanding regarding another LLM evaluation. As promised, we have now added preliminary results for the new model, Ministral-3b-instruct, to Appendix H of the revised paper. We will continue this experiment with the full dataset and include the results in the camera-ready version.
>
> - Regarding the parameter c, we agree with your suggestion to include a summary of our observations. This has been addressed in the revised version (Appendix D).
>
> - To address your concern about prediction time, we have extended Table 1 in the revised paper to include inference time for batch size 1 on both CPU and GPU.

---

> > ### Comment · Reviewer_bvWq · 2024-12-02
> >
> > Thanks for adding the inference time for a batch size of 1, but could you please explain how this time is measured?
> > I would have expected an inference time per sample (TPS) a bit higher than the one for batches of 512 since any overhead (e.g., memory to the GPU, etc.) should now be only attributed to a single sample rather than averaged across all samples of one batch. Instead, the inference times per sample for batch size 1 reported in Table 1 are smaller than the ones for all other batch sizes.

---

> > > ### Author Response · Authors · 2024-12-02
> > >
> > > Thank you for your observation. The inference times for a batch size of 1 in Table 1 were reported in milliseconds, which we should have specified explicitly. For consistency with other entries in the table, we will update the table to reflect these values in microseconds. The correct values are 155.2946945336693 microseconds (or 0.155 ms) and 87.43007441425092 microseconds (or 0.087 ms).
> > >
> > > Regarding the measurement methodology, we measure the total time required to complete the entire evaluation workload for each <device, batch size> setting and divide this total time by the number of samples to compute the time-per-sample. Each <device, batch size> setting is evaluated 20 times, and we report the mean and standard deviation of the TPS values. All time measurements are conducted on the same testbed.
> > >
> > > We hope this clarifies the reported numbers and our measurement approach.

---

### Official Review · Reviewer_DYK6 · 2024-11-03

**Soundness:** 3
**Presentation:** 4
**Contribution:** 3
**Rating:** 6
**Confidence:** 4

**Summary:**

In this work, the authors propose a strategy for predicting output length that builds upon state-of-the-art approaches such as S3. The approach modifies output length prediction by leveraging embeddings from different layers of the LLM and providing them to a trainable, lightweight predictor module. These predictions are then used in conjunction with size-based scheduling approaches such as SRPT, resulting in reduced overall latency and TTFT. The key idea here is that length-aware scheduling reduces memory overhead and, consequently, preemptions.

**Strengths:**

A major strength of this paper is its output length prediction module, which iteratively refines predictions using embeddings from the LLMs. This approach enables the use of SRPT-based scheduling methods. The work also proposes two approaches for predicting output length and discusses the overhead associated with LLM inference. Additionally, the paper presents closed-form expressions based on queuing theory for mean response time.

**Weaknesses:**

The paper has a few weaknesses that should be considered.
1. The output length prediction assumes uniform bucket sizes, which may lead to overfitting if the number of requests in each bucket varies significantly. A comprehensive evaluation of the distribution per bucket is required for a fair evaluation of this approach.
2. The assumption that the output length predictor needs access to the layers of the LLMs may not hold for many OpenAI models, whose internal architecture is unknown. The generalizability of the approach in such cases needs to be discussed.
3. Finally, some tasks from the Alpaca dataset are more unpredictable than others. For instance, tasks such as writing a story are far more arbitrary than classifying a sentence. Therefore, remarks on the performance of the approach with respect to different tasks should be added for better generalizability.
4. It is unclear whether a certain layer carries most of the information about the output length for every LLM, or if this is a generic assumption that may not hold for all models

**Questions:**

1. Can the generalizability of this approach across different LLMs be evaluated? Is it a generic assumption that for every LLM, a certain layer carries most of the information about the output length, or could this vary depending on the model?
2. How can this approach be applied in the cases of OpenAI models where the internal architecture is unknown?
3. What is the output length distribution used in the paper, and how would the performance vary if the buckets are selected based on their size?
4. Can the paper provide more information on the Alpaca dataset, and what was the distribution of the data used with respect to the root verb of instructions from the Alpaca dataset?
5. How does the paper ensure that the performance of the approach is consistent across different tasks within the Alpaca dataset, given that some tasks may be more unpredictable than others? Can the authors provide the breakdown of the MAE and heatmap with respect to different root verbs of instructions?

---

> ### Author Response · Authors · 2024-11-23
>
> Thank you for supporting our approach and recognizing the contributions of the output length prediction module and the theoretical closed-form equation for the mean response time of the proposed scheduling policy.
>
> - Q1: Can the generalizability of this approach across different LLMs be evaluated? Is it a generic assumption that for every LLM, a certain layer carries most of the information about the output length, or could this vary depending on the model?
>
> A1: We appreciate your comment regarding another LLM evaluation. We have now added preliminary results for the new model, Ministral-3b-instruct, to Appendix H of the revised paper. We will continue this experiment with the full dataset and include the results in the camera-ready version (due to time and resource constraints during the rebuttal period). We see that the assumption that certain layers in LLMs carry output length information also holds for the newly evaluated model, as these architectures share foundational design principles.
>
> - Q2: How can this approach be applied in the cases of OpenAI models where the internal architecture is unknown?
>
> A2: While our work assumes access to the model's internal layers, the proposed scheduling policy can still work with black-box models, provided output size predictions are available. For OpenAI models, if the model can return the remaining token count alongside each generated token, our approach, TRAIL, could be directly applied. Even without this, our scheduling policy, which limits preemption based on external output predictions, can still be effective if some other external prediction is available. Analyzing scheduling for black-box models is an interesting direction for future work.
>
> - Q3, Q4, Q5: More information on the Alpaca dataset, output length distribution, and consistency of performance across tasks with varying unpredictability.
>
> We appreciate the reviewer’s concerns. In the revised paper, we have added the output length distribution and bins histogram (Appendix F).
> We acknowledge that uniform bin sizes may lead to biases, and we followed the same settings used in $S^3$[1], which also uses the Alpaca dataset. For the camera-ready version, we will explore non-uniform bin sizes (e.g., bins of geometrically increasing size such as 2, 4, 8, …) to address this.
> Regarding the Alpaca dataset, while we considered breaking the data based on root verbs, we believe comparing results across datasets would provide a clearer evaluation. Due to time and resource constraints, we could not perform this comparison during the rebuttal but aim to include an additional dataset in the camera-ready version.
>
> [1] $S^3$: Increasing GPU Utilization during Generative Inference for Higher Throughput, Jin, Yunho, Neurips 2023.

---

> > ### Comment · Reviewer_DYK6 · 2024-12-02
> >
> > Thanks for addressing the comments. I prefer to retain the previous rating.

---

### Official Review · Reviewer_KSCH · 2024-11-07

**Soundness:** 2
**Presentation:** 3
**Contribution:** 3
**Rating:** 6
**Confidence:** 3

**Summary:**

This paper introduces TRAIL, a method for predicting the remaining output length of an LLM based on its internal structural embeddings. Using these predictions, the authors propose a scheduling algorithm, a variant of the shortest remaining processing time, to reduce latency. They derive the theoretical expected response time for a job and implement the scheduling policy within the vLLM serving system for evaluation.

**Strengths:**

The problem the authors address: the inefficiency of FCFS scheduling in LLM serving, is practical. Overall, I think the proposed solution is useful. Online token length prediction is inherently challenging, and their approach of using bins and classifiers is reasonable based on my experience. The feature design for the predictor is a novel contribution. The paper appropriately tackles key considerations in LLM serving, such as the uncertainty in output token length, the difficulty of making accurate predictions, the drawbacks of adding memory overhead during runtime decoding, and the distinction between the initial token generation (prefill) and the decoding of subsequent tokens.

**Weaknesses:**

Some terms could be more precise and consistent. For example, in the context of LLM serving, what the paper refers to as a “job” is actually a “request.” The term “job” typically implies a collection of multiple requests or tasks. While the authors note that “job” and “request” are used interchangeably, this distinction isn’t necessary and could lead to confusion. Additionally, the term “TRAIL” is described as a prediction method in the abstract but appears to refer to a scheduling policy in the evaluation section. This inconsistency should be clarified.

In Figure 6, the main experimental results show that the trends for Trail and Trail-BERT exhibit a faster growth rate compared to vLLM-FCFS and vLLM-SJF_BERT. Could you increase the request rate further to demonstrate whether the latency of Trail and Trail-BERT will remain below that of vLLM-FCFS and vLLM-SJF_BERT as the request rate continues to increase?

In multi-GPU settings with larger models, the results may differ due to the additional time required for initializing the Ray engine, GPU-to-GPU communication, and other overheads. While I believe the scheduling algorithm remains effective, it would be beneficial to evaluate its performance on larger models in a multi-GPU environment.

I'll raise the score if the authors successfully address the experiment issue.

**Questions:**

Could you explain why the performances of vLLM-FCFS and vLLM-BERT are so close in all metrics? Is it because that BERT prediction does not work at all, as shown in figure 3?

---

> ### Author Response · Authors · 2024-11-23
>
> We thank the reviewer for the thoughtful feedback and valuable suggestions, which have helped us improve the clarity and quality of our work.
>
> - We appreciate the suggestion to extend the evaluation:
>
>    - In the revised paper, we have considered increased request rates and updated Figure 6 accordingly. As predicted, Trail continued to exhibit benefits at higher request rates due to the mitigation of head-of-line blocking, which is consistent with queueing theory predictions. We also added to Figure 6 Trail+, a baseline oracle where requests are scheduled based on their exact length.
>
>
>   - For testing multi-GPU settings, we used a machine with dual AMD EPYC 7313 CPUs (16 cores per CPU, totaling 64 threads), 503 GB of RAM, and two NVIDIA A100 GPUs with 80 GB memory each connected via NVLink. We evaluated our approach in multi-GPU settings using two methods:
>
>       1- Distributing the existing tested model across two GPUs. The results in Figure 11, Appendix G, show that TRAIL continues to perform effectively.
>
>     2- Since training our classifier for larger models takes resources and time (While we aim to profile and train our classifier for these larger models for the camera-ready version), we tested TRAIL with a larger model size (vicuna 13B), using "perfect" predictions (denoted as Trail+) using sampling 10000 prompts from the first 1000 prompts from Alpaca dataset. In the camera-ready, we will complete all baselines and the full dataset. Trail+ serves as an upper-bound benchmark for this scenario. These results appear in Figure 12 Appendix G.
>
>
> - We acknowledge the reviewer’s concern regarding using both the terms request and job. We clarified the description of TRAIL throughout the paper and revised the paper to consistently use "request" in the context of LLM to avoid confusion. We do note that in queueing “job” is commonly used (as in the Shortest Job First policy), but we believe this change provides helpful clarification.
>
> - Q: “Could you explain why the performances of vLLM-FCFS and vLLM-BERT are so close in all metrics?”
>
> A: vLLM-SJF_BERT sticks to vLLM's implementation, where incoming requests are prioritized. For the remaining slots, SJF is applied using BERT predictions. This approach adjusts the ordering policy but does not modify the underlying scheduling mechanism of vLLM. Comparing BERT predictions with our approach, we compare TRAIL-BERT with TRAIL in Figure 6.

---

> ### Author Response · Authors · 2024-12-02
>
> Dear Reviewer KSCH,
>
> Thank you once again for your thoughtful feedback. We appreciate the time and effort you spend reviewing our work.
>
> As the ICLR public discussion phase draws to a close, we wanted to confirm whether our responses properly addressed your concerns. If there are any remaining questions or points requiring further clarification, please let us know—we would be glad to provide additional details.

---

### Official Review · Reviewer_uAxn · 2024-11-09

**Soundness:** 3
**Presentation:** 3
**Contribution:** 3
**Rating:** 8
**Confidence:** 4

**Summary:**

The paper addresses the challenges of efficient scheduling in interactive Large Language Model (LLM) applications. It uses a combination of length prediction and batching for speedup.  It introduces TRAIL, a method that uses the LLM’s own embeddings to predict the remaining length of requests, enabling a prediction-based Shortest Remaining Process Time (SRPT) variant with limited preemption. This approach aims to reduce memory overhead and optimize resource utilization by allowing preemption early in request execution and restricting it as requests near completion. Experiments show lower mean latency and time to the first token compared to current approaches.

**Strengths:**

1. The use of LLM embeddings for predicting residual request lengths, is novel, interesting, and intuitive. It has a clear potential to improve request scheduling as seen in the evaluation in this paper and can also be useful in any scenario where output length prediction is required and thus can have a broad positive impact.

2. The use of pre-emption based on length prediction has also not been fully explored in other works to the best of my knowledge. The authors propose a clear SRPT-like approach based on predicted request length and also make interesting observations about the extent of pre-emption that is useful (Fig 5) which will be useful for future research in this space.

**Weaknesses:**

1. The main algorithm is not very clearly explained although the issues are relatively minor and can be fixed by addressing questions 1-4 below.

2. By solely relying on length-based scheduling there is a risk of violating latency SLAs. In general, a real-world scheduler should be able to tune between FCFS and length-based scheduling to handle requests with different SLAs.

3. The evaluation is largely limited to the mean latency and does not adequately capture the tail behavior of the proposed approach or baselines. I would recommend including the corresponding P95 latency plots, along with comments, for Fig 5,6, and 7 to address this.

**Questions:**

1. It seems strange to take the average of prefill embeddings for inference. Is this the only approach you have tried, or did you try other (for e.g. non-linear) ways to aggregate the prefill embeddings?

2. What is 44? (shape of the embedding tensor after the prefill phase is [1,44,4096] as mentioned in Section 3.1 on page 4)

3. What is $\hat{q}_\text{prior}$? How is it initialized?

4. What is the LLM forward pass latency per token for the setting(s) in Table 1? Please include that number so we can get a clearer idea of the latency overhead of length prediction.

5. There appears to be an inconsistency in notation between 'C' in Section 3.3 and 'c' in Section 4.2. Please clarify if they are the same or different and fix the notation accordingly. Also do you have any ideas on how one could choose a good value of 'C'/'c' in real world? Can it be learned/adjusted in real-time based on the batch/request characteristics?

6. It is mentioned in Section 4.2 that vLLM-SJF_BERT prioritizes incoming requests over existing running requests. What does that mean? Shouldn't it be scheduling requests just based on the predicted length and not based on arrival time?

7. I would recommend adding an oracle baseline where requests are scheduled based on their exact decode length. While the true decode length will not be known in practice, this will serve as a useful benchmark and approaches can be ranked based on how close they are to this one.

---

> ### Author Response · Authors · 2024-11-23
>
> We thank the reviewer for supporting our paper. We appreciate the thoughtful questions that clarify essential details of our approach, as well as your pointing out typos. In the revised paper, we have added to Figure 6 a new baseline  Trail+, which shows results for an oracle that provides exact lengths and schedules accordingly. We have also extended Figure 6  with increased request rates. As predicted, Trail continued to exhibit benefits at higher request rates due to the mitigation of head-of-line blocking, which is consistent with queueing theory predictions. We also added LLM forward pass latency per token in Table 1.
>
> - Q1: “Is this the only approach you have tried, or did you try other (for e.g. non-linear) ways to aggregate the prefill embeddings?”
>
> We used averaging as a simple and efficient baseline that does not depend on input size to demonstrate the feasibility of using embeddings for length prediction. While we did not explore non-linear methods in this work, we agree they could improve accuracy and plan to investigate them in future research. We aimed to establish the foundation and demonstrate the utility of embedded-informed scheduling.
>
>
> - Q2: “What is 44?”
>
> Thank you for pointing this out. The value "44" was a mistake in the original text and represents the number of input tokens. The correct parameter list should be (1, [input tokens], 4096), where "44" is just an example of the number of input tokens. This clarification has been added to the revised paper.
>
>
> - Q3: “What is $q_{\mathrm{prior}}$​? How is it initialized?”
>
> $q_{\mathrm{prior}}$ represents the prior probability estimate of refined predictions at iteration t during Bayesian inference. At t=0, it​ is initialized to the initial prediction p(0). We have fixed the notation in the revised paper.
>
>
> - Q4: “What is the LLM forward pass latency per token for the setting(s) in Table 1?”
>
> We added LLM forward pass latency per token to Table 1 to clarify the prediction overhead in our tested model.
>
>
> - Q5: “There appears to be an inconsistency in notation between 'C' in Section 3.3 and 'c' in Section 4.2.”
>
> This is a typo; thank you for pointing to this. Both notations refer to the same parameter. We have corrected this in the revised version.
>
> - “How to choose a good value of 'c' in real-world scenarios? Can it be learned/adjusted in real-time based on batch/request characteristics?”
>
> The value of 'c' primarily depends on the available memory for the KV cache, which is influenced by model size, batch sizes, and incoming request sizes. For instance, preempting long requests could monopolize memory and reduce throughput during periods with long requests followed by short ones before finishing. As shown in Appendix D (Figure 8), workload simulations can help determine suitable 'c' values, which can also be adjusted in real time without further system changes. This adjustment directly impacts the rank of jobs (see Equation 1 in Appendix C), connecting the 'c' value to the request ranking. This discussion was added to Appendix D as well.
>
>
> - Q6: “It is mentioned in Section 4.2 that vLLM-SJF_BERT prioritizes incoming requests over existing running requests. What does that mean? Shouldn't it be scheduling requests just based on the predicted length and not based on arrival time?”
>
> vLLM-SJF_BERT sticks to vLLM's implementation, where incoming requests are prioritized. For the remaining slots, SJF is applied using BERT predictions. This approach adjusts the ordering policy but does not modify the underlying scheduling mechanism of vLLM.
>
>
> - Q7 “I would recommend adding an oracle baseline where requests are scheduled based on their exact decode length.”
>
> We agree and added this Oracle baseline (TRAIL+)  to Figure 6 in the revised version.
>
>
> - We understand the concern about the risk of violating latency SLAs. Our intuition from previous work on queueing is the benefits of prioritizing jobs by length (or predicted length) provide such significant gains in expected latency that the system can be modified to handle tail latency issues while maintaining much of these gains.  In particular, one way to address this is by adding a starvation prevention mechanism. We will add this to the camera ready and evaluate the tail latency under different parameters.

---

> > ### Comment · Reviewer_uAxn · 2024-11-28
> > **Re**
> >
> > Thank you for addressing my concerns. I believe this paper is ready for publication and have increased my score to reflect the same. I would still recommend including plots on the tail latency in the camera-ready version but otherwise the paper in its current form looks good to me.

---

### Author Response · Authors · 2024-11-23

We thank the reviewers for their thoughtful feedback and are encouraged by their recognition of our work. We appreciate that they found the problem practical (KSCH) and our approach—leveraging LLM embeddings to predict request lengths— novel, interesting, and intuitive (uAxn, KSCH, DYK6). Reviewers also highlighted the clear potential of our approach to improving request scheduling (uAxn, bvWq) while being much simpler than existing state-of-the-art methods (bvWq). Additionally, they acknowledged that this is the first work to address the memory overhead of preemption scheduling policies (uAxn), which they view as valuable for future research in this space (uAxn).
We are particularly pleased that reviewers (DYK6, bvWq) appreciated the contribution of closed-form expressions derived from queueing theory to compute the mean response time for our proposed scheduling policy, noting that these insights have not been explored previously in queueing theory. Reviewer uAxn also highlighted the broader potential impact of our approach, both in the context of output prediction and in advancing research on preemptive scheduling policies where there is memory limitation (decode phase in LLM, for example).

In the revised paper, we made the following updates:

- Extended the evaluation to consider increased request rates. (Figure 6)

- Added Trail+, a baseline oracle where requests are scheduled based on their exact length. (Figure 6)

- Included LLM forward pass latency per token and inference time of batch size 1. (Table 1)

- Evaluated our approach in a distributed multi-GPU setting. (Figures 11, 12 Appendix G)

- Added the output length distribution and bins histogram (Appendix F).

- Added preliminary prediction results for a new LLM model, Ministral-3b-instruct (Appendix H).

We address reviewer comments below and will incorporate all feedback.

---

### Comment · Area_Chair_H1yW · 2024-11-25

Dear reviewers,

As the deadline for discussion is ending soon. Please respond to the authors to indicate you have read their rebuttal. If you have more questions, now is the time to ask.

AC

---

### Meta-Review · Area_Chair_H1yW · 2024-12-19

**Metareview:**

This paper introduces TRAIL, an approach to efficient scheduling in interactive LLM applications by combining length prediction and batching for improved performance. The work demonstrates several contributions: it presents an innovative use of LLM embeddings for predicting residual request lengths, which shows clear potential for improving request scheduling. The problem being addressed - the inefficiency of FCFS scheduling in LLM serving - is highly practical, and the proposed solution effectively tackles key considerations in LLM serving. The output length prediction module, which iteratively refines predictions using embeddings from the LLMs, is particularly noteworthy. The work achieves significantly lower mean latency compared to baselines and offers a theoretical proof of the proposed scheduling scheme.

While the paper presents compelling contributions, reviewers identified several areas for improvement. The main algorithm requires clearer explanation (Reviewer uAxn), and there are concerns about the potential violation of latency SLAs when relying solely on length-based scheduling (Reviewer uAxn). The evaluation would benefit from including P95 latency plots to better capture tail behavior (Reviewer uAxn). Some terminology could be more precise and consistent, particularly regarding the use of "job" versus "request" (Reviewer KSCH). The output length prediction assumes uniform bucket sizes, which may lead to overfitting (Reviewer DYK6), and the approach's generalizability across different LLMs needs further exploration (Reviewers DYK6, bvWq). Additionally, a deeper sensitivity analysis of key parameters would strengthen the work (Reviewer bvWq). Despite these limitations, the paper's novel contributions and practical significance lead to acceptance.

**Additional Comments On Reviewer Discussion:**

All reviewers are satisfied with the rebuttal and some even increased the score.

---

### Decision · Program_Chairs · 2025-01-22

Accept (Poster)